# Reconciling the God of Traditional Theism with the World's Evils

## Robin Attfield 

School of English, Communication and Philosophy, Cardiff University, Cardiff CF10 3BA, UK;
attfieldr@cardiff.ac.uk

**Abstract:** Replying to James Sterba's argument for the incompatibility of the world's evils with the existence of the God of traditional theism, I argue for their compatibility, using the proposition that God has reasons for permitting these evils. Developing this case involves appeal to an enlarged version of both the Free Will Defence and Hick's Vale of Soul-Making Defence, in the context of God's decision to generate the kind of natural regularities conducive to the evolution of a range of creatures, including free and rational ones. Sterba writes as if God would be required to authorise frequent infringements of these regularities. Sterba's arguments from ethics and from the inadequacy of post-mortem compensation are problematised. Predicates used of God must bear a sense appropriate to the level of creator, and not of a very powerful cosmic observer. The ethics that applies within creation should not be confused with the ethics of creating.

**Keywords:** moral evil; natural evil; Free Will Defence; laws of nature; miracles; James P. Sterba

## 1. An Argument for the Compatibility of God and the World's Evils

James Sterba has presented arguments (both in a book and in an article) intended to show that the existence of the God of traditional theism (who is understood to be omnipotent and all-good) is incompatible with the evils of the actual world. Hence, there can be no such God (Sterba 2019, 2020).

While there is insufficient space to reply to all of Sterba's arguments (a book-length endeavour in itself), I would like to reply here to some central and crucial ones. Thus, I hope to present at least the makings of an approach that can reconcile the God of traditional theism with the world's evils.

I will begin by ventilating an argument for the compatibility of the existence of the God of traditional theism and the evils present in the actual world. A standard way to establish that two distinct propositions (A and B) are consistent, and thus their compatibility with each other, is to find and specify that a third proposition which is consistent with A is possibly true, and which, in conjunction with A, implies B. This procedure has been outlined by, for example, Stephen Davis in his own essay in his collection *Encountering Evil* (Davis 1981).

So let us consider, by way of a relevant additional proposition, the claim that God has reasons unknown to us for creating a world having all the evils of the actual world. This claim (let's call it "C") certainly seems consistent with the existence of the God of traditional theism (a proposition which for present purposes we can designate as "A"); it appears to be possibly true; and (whether in conjunction with A or not) it implies the existence of all the evils of the actual world (a proposition which we can designate as "B"). Accordingly, this argument appears to show the consistency of A and B, and thus the compatibility of the existence of the God of traditional theism with that of the evils of the actual world.

Now C would be rather unsatisfactory if the argument were intended to persuade others of the probability of God's existence, because of its appeal to unknown reasons. There again, I am unsure whether it is actually true, since at least some of God's reasons might be known to, or at least grasped by, some of us. But none of this matters for our present purposes. For all that is needed (apart from C

being consistent with A, which is unproblematic, and implying B, which similarly raises no problems) is that it is possibly true, or, in other words, not necessarily false. And that is just what it appears to be.

Sterba could, however, question this claim about C. God, he might claim, being omnipotent and all-good, could not have reasons (known or unknown to us) for creating all the evils of the actual world. This is because an all-good God with powers of omnipotence would prevent some of these evils, and would therefore not create a world in which they were present (Sterba 2019, pp. 71–97), nor have reasons to do so. This reply must also, to serve its function of rendering the argument for compatibility unsuccessful, claim that C is not even possibly true, but is necessarily false. This may seem rather a heroic move; but it is still apparently a possible one.

Now the proposition that serves the role of C is not the only one that has been proposed for this role in arguments for the compatibility of God's existence and the evils of the actual world. Thus, Davis has proposed a different one (Davis 1981, p. 72), and I a different one again, and one that is slightly more plausible at that (Attfield 2006, p. 135). But in each case, Sterba could attempt to make a parallel move, provided that in each case the additional proposition refers to God as its subject. And if it does not refer to God, but still implies the existence of the evils of the actual world, then he could claim that there is the same inconsistency between the pair comprising it and A as he purports to find between the pair comprising A and B.

However, the imagined reply to the above compatibility argument, namely that God could not have reasons, known or unknown, for creating a world having the evils present in the actual world, can itself be criticised by presenting reasons that God might have. This kind of reply, of course, could be held to concede that not all of God's reasons are unknown to us. Yet if some possible reasons can be presented, this can be done without any claim to know fully what God's reasons are. For there being possible reasons that could serve to reconcile actual evils with God's existence would suffice to overthrow the claim that God's existence and the evils of the actual world are incompatible, and also the claim that God could not have reasons (of any sort) for creating a world having all the evils of the actual world.

## 2. A Possible Reason for God's Creation of Our World

One of the reasons that God could have for creating a world with evils like those of the actual world is the value of free will and its implications, such as freely chosen actions and the freedom of thought that it presupposes. A world containing creatures with this kind of freedom can be held to be much more valuable than one without such creatures, even if some of their choices are morally wrong, and even if some of their freely chosen actions have evil consequences. This line of thinking gives rise to the "Free Will Defence", a sophisticated version of which has been presented by Alvin Plantinga (Plantinga 1965, 1974).

Sterba, however, contests this defence in a chapter called "There is No Free-Will Defense" (Sterba 2019, pp. 11–34). According to Sterba, God has failed to promote (let alone to maximise) the kind of freedoms that would be protected by a just state, freedoms including access to resources needed for human flourishing; and some of these freedoms are more significant than the freedoms, for example, of assailants to injure, maim or kill their victims (which the free will defence regards God as upholding). While I do not altogether endorse Sterba's version of political libertarianism, there is no need to explore the relevant reservations here. For, even if the evils that befall the victims of assailants or the poor whose lack of access to resources prevents their flourishing are understood as absences of welfare or well-being rather than as absences of freedom, a case parallel to Sterba's could be mounted that a good God would have prevented these evils, and promoted their flourishing in that way, rather than promoting the freedoms which the Free Will Defence turns on. Indeed, Sterba maintains that "there is no free-will defence" at all, since God does not intervene to prevent the violation of what Sterba regards as significant freedoms.

But this argument neglects the differences between the situation of a creator and of moral agents such as ourselves, or such as governments or the state; and it also neglects the crucial importance of

freedom of the will (which some call "metaphysical freedom"). Let us consider the second point first. Those who deploy the Free Will Defence are not seeking to explain how or whether God promotes personal or political freedoms. They are instead concerned to explain why a good God might permit those evils that result from human choices (and the free choices of other animals, if there are any such free choices). Here, like Sterba, I am assuming an incompatibilist understanding of freedom, rather than a compatibilist one. The freedom to choose is fundamental to the kinds of action characteristic of human life and culture, and without it human life would lack autonomy, making human beings robots or puppets governed by their genes and their environment. Other personal and political freedoms are for human agents and societies to promote and defend, through the exercise of this basic freedom. And since it sometimes results in evils, God's permitting of these evils can largely be justified by the significance of this kind of freedom, and relatedly his or her creation of creatures equipped with it.

Another way of expressing the distinction between the freedoms of Sterba's argument and the freedom which is the focus of the Free Will Defence has been presented in a review of Sterba's book by Michael Almeida. To be significantly free is to have metaphysical freedom of this kind, freedom of choice that can be exercised in a wide range of moral situations. The freedoms that Sterba emphasises, such as political freedoms, are politically significant, and ones that the state should defend, but they lack the crucial significance that attaches to metaphysical freedom. Besides (as Almeida proceeds to show), these two kinds of freedom are independent of each other. "We can have significant freedom and possess no political freedoms at all. Further we can possess every political freedom and totally lack metaphysical freedom." (Almeida 2020) This latter claim needs to be qualified, because someone so brain-damaged as to lack metaphysical freedom would lack some political freedoms as well (such as the freedom to vote) in all practical respects, even if still possessed of this freedom in abstract theory. Yet it remains that a good God might decide to create a world containing creatures equipped with metaphysical freedom while not interfering with respect to political freedoms and leaving the widespread attainment of these freedoms to human agents (with all their created capacities) and their political societies.

Here we should return to the matter of some of the differences between the situation of a creator and that (or those) of moral agents. The sheer breadth of the powers of an omnipotent creator means, as Almeida remarks, that if God were to preclude certain wrong actions, then we would all be rendered unable to perform them. But this would arguably prevent the valuable actions and valuable omissions of most moral agents, who freely refrain from performing these wrong actions. Just (political) states, as he adds, "should initiate policies to prevent such evils, because, unlike divine policies, doing so would not seriously diminish the moral value of the world" (Almeida 2020). Further, I would add, the legislation of just states would not change the entire order of the world across space and time, which is precisely what a divine decree would be liable to do. But this is a theme to which I will return below in a different context.

Accordingly, the Free Will Defence has a role to play in a reconciliation of God's power and goodness and the world's evils (or theodicy), because it explains how significant freedoms, together with many of the evils that they give rise to, are facilitated by the creator. It does not completely explain all the evils that result from human choices, for it does not explain, without supplementation, why God does not interfere to prevent the most horrendous of evils resulting from free but immoral actions. However, together with supplementary explanations (see below), it performs a key role in any acceptable theodicy.

## 3. Hick's Irenaean Theodicy

There is another theodicy, regarded by its author, John Hick, as a replacement of the Free Will Defence, but in my view complementary to it. (Hick also supplies a defence of the Free will Defence against critics such as Mackie (Mackie 1955; Hick 1977, pp. 266–77), but still finds it theologically defective (Hick 1977, pp. 277–80).) Hick considers its origin as in some part due to Irenaeus (c. 130-c. 202), who held that humanity was first created morally immature and needed to gain in

maturity and also to attain in each generation the full stature of humanity and moral character (Hick 1977, pp. 211–15, 253–55). A major strength of Hick's adaptation of this approach lay in his relating it to evolution and to the development of humanity through descent from pre-human ancestors (Hick 1977, pp. 280–87). Thus, human development involved overcoming difficulties and temptations, rather as the life of human infants does on the way to the maturity of adulthood. Accordingly, a world affording unstinted pleasure or unmitigated happiness was inappropriate to the development of moral character, or, as Hick names it, "soul-making". Had he been aware of Sterba's book, he might have added that a world in which moral and political freedoms were maximised would have been equally inappropriate.

Sterba is aware of this theodicy, but regards it as not explaining the worst outcomes of free human action, any more than the Free Will Defence does. For many of the sufferings and injuries inflicted by humans upon their fellows arguably impede the development of moral character, or, in the case of the infliction of premature death, prevent it, at least in this life. Hick, for his part, held that the process of moral development can continue into the next life, and Sterba spends some of his book discussing such matters (Sterba 2019, pp. 35–48). In any case, a theodicy centred on "soul-making" appears to need to be supplemented with some further theodicy if God's permitting the worst outcomes of free human action is to be explained.

Nevertheless, the Irenaean theodicy could be held to explain many other aspects of human existence and of the world around us. For the development of moral character appears to require an environment of dangers and difficulties, which each generation has to learn either to surmount or to live with. The development of character also presupposes a journey or trajectory along many forking paths, each of the alternative pathways appearing attractive to one or another element in the blend of desires that we inherit from our pre-human ancestors. Mary Midgley has well explained how such inherited drives as the desire to protect offspring and the potentially conflicting desire to take flight or to join migrations together with conspecifics, drives supplied by our evolutionary ancestry, actually made freedom of choice possible among incipiently rational early human beings (Midgley 1994, pp. 160–61). Such dilemmas are needed for the development of moral character in every generation.

Hick himself contributed another strand to such a theodicy. God's desire for human beings freely to choose to enter into a relationship with him or her, and freely to take the path of right action, required the correctness of these pathways not to be clear and manifest, but to be hedged about with doubts and hesitations. Thus, in keeping with this desire, God would be likely to install a cognitive or epistemic distance between humanity and God. Hence, the principles of right action are obscure, and belief in God's very existence is far from obvious. This divine policy could even be held to dovetail nicely with the current debate about the compatibility of God's existence and the world's evils, where the truth is particularly difficult to discern.

Thus, Hick's Irenaean theodicy helps explain how an all-good God might decide to create a world with living creatures undergoing an evolutionary process, both for their own good and for that of the human beings of which such evolved creatures were the ancestors. It can explain many of our epistemological problems, as we learn to distinguish reality from illusions, and our moral problems, as we learn to distinguish right from wrong. It can even help to explain the very development of our capacities for free choice, emphasised in the Free Will Defence, and many of the bad and often disastrous consequences of our choices, and of God's permitting them to come about; for their prevention would frequently block the route to the development of moral character.

Yet this theodicy does not explain God's permitting atrocities, or, in Sterba's phrase, the "horrendous evils" that human action sometimes generates, any more than the Free Will Defence does. Or, at least, it does not do so without a considerable expansion in our awareness of the kind of framework that the "soul-making" theodicy involves. How to expand this theodicy and the Free Will Defence will be considered in Section 5, after a brief discussion of some ethical principles in Section 4.

### 4. The Pauline Principle and the Principle of Double Effect

The Pauline principle is expressed in Paul's Epistle to the Romans, 3:8: Do not "do evil that good may come". But this form of words needs to be disambiguated. One possible meaning is "Do not do wrong that good may come"; but the people who attributed "do evil that good may come" to Paul could not have meant this question-begging command. They (and Paul in his response) must rather have meant (Do not) "cause bad things to happen that good may come (of it)".

But, as Almeida says, this principle cries out for qualification, as Sterba himself acknowledges. For Sterba acknowledges that it could be right, as "the only way to prevent a far greater harm to innocent people" to "shoot one of twenty innocent hostages to prevent, in the only way possible, the execution of all twenty" (Sterba 2019, p. 50).

Yet there is a vastly wider range of cases than this particular one (devised originally by Bernard Williams) where it may be right to cause bad things to happen. Thus, it may often be right to punish children and animals to secure better behaviour. Farmers believe that it can be right to castrate rams and billy-goats so as to rear them better, and, while they may form this view with too little reflection, it is implausible to claim that they are invariably wrong about this. Those of us who believe that living creatures are moral patients, and thus matter, morally speaking, still consider it right to trim hedges for the sake of the general good of a garden. And to come closer to the treatment of human beings, all states appear to believe that it can be right to coerce people from driving on the wrong side of the road, and to punish both citizens and other residents so as to prevent infringements of the law, and promote the law-abidingness of society.

A principle requiring such a wide range of exceptions for human agents can hardly be one that is mandatory for God if God is to count as all-good. Sterba considers that the permissible exceptions "allow us to do evil that good may come of it only when the evil is trivial, easily reparable, or the only way to prevent a far greater harm to innocents". He concludes that "it is difficult to see how God's widespread permission of the harmful consequences of significantly evil actions could be a justified exception to the Pauline Principle" (Sterba 2019, p. 50). But the Principle can be held to require a much ampler range of exceptions. Thus, coercive actions that prevent harm to non-innocents as well as to innocents are assumed to be widely permissible in law. Trials do not take place to assess the moral guilt or innocence of possible victims, before perpetrators of violent crimes are deemed eligible for punishment. More significantly, acts of significant violence are widely considered justifiable in a justified war, a relevant example being when the war is fought to overthrow tyranny or to establish a just peace—or, in other words, that good may come.

The Pauline principle, then, admits of too many exceptions to show that God should intervene to prevent evils. A slightly more presentable principle could instead be considered, the Principle of Double Effect, a principle that makes agents morally responsible for the intended consequences of their actions, but not for the foreseeable ones. If such a distinction can be made in the case of God, then someone might imaginably argue that God should not permit evils, even when he or she can foresee a range of good which could result, or for which the evils were necessary conditions. But Sterba actually rejects the distinction between intended and foreseeable consequences in the case of God (Sterba 2019, pp. 51, 67, n. 4), holding perhaps that God's power and knowledge are such that consequences that God foresees as emerging from actions or omissions that he or she intends are just as much intended as what are understood as God's intentional actions and omissions. Accordingly, Sterba does not make appeal to the Principle of Double Effect. It could be replied to this stance that just as human agents need not be held to intend the bad states of affairs that they bring about with a view to producing good outcomes, or as necessary conditions of good outcomes, no more should God be held to intend the bad outcomes of events and actions that she or he permits with a view to these events and actions making good outcomes possible. Accordingly, the distinction between intended and foreseeable consequences applies as much to God as to human agents.

Here we need to make a distinction between God's antecedent will and God's consequent will, a distinction that has been well drawn by Keith Ward (Ward 2007, pp. 48–55). Antecedently, God



wills there to be a world with its system of laws and its array of creatures, some of them rational and equipped with free will. The actions of these creatures, and their (often undesirable) outcomes, are dependent or consequent on God's will, but many of them need not be regarded as wanted by God. As Ward puts matters: " . . . one can certainly say that many things can happen in a created order, wholly dependent for its existence upon God, which God does not intend, desire or approve of" (Ward 2007, p. 55).

But none of this makes the Principle of Double Effect a suitable basis for arguments against (or indeed for) belief in God's goodness. For there are numerous strong objections to this principle, even with regard to human agents. I have set out some of these objections in *A Theory of Value and Obligation* (Attfield 2020, pp. 129–32), and in *Value, Obligation and Meta-Ethics* (Attfield 2019, pp. 128–31); this is not the place to repeat them. This Principle has also been criticised by Judith Lichtenberg (Lichtenberg 1994). Thus, the Double Effect Principle is morally dubious itself, even when applied to contexts relating to human action, let alone to that of divine action. It is time to turn instead to a broader understanding of the context in which both the Free Will Defence and Hick's Irenaean theodicy may be taken to operate.

## 5. The Broader Context: Cosmic Regularities

Sterba holds that an all-good God would intervene to prevent the significantly bad impacts of freely chosen human actions. He recognises that if God were to intervene in this manner, then the world would not be regular in the way in which the world actually is. As he puts this objection:

> Still, it might be objected that if God did intervene to the degree to which I am claiming he would have to be intervening, we would no longer be living in a world governed by natural laws, and so no longer able to discover such laws and put that knowledge to work in our lives.

But he at once presents a reply to this objection:

> Clearly, there is no denying that a world where God intervened, as needed, to prevent significant and especially horrendous evil consequences of immoral actions would be a different world from the one we currently inhabit. But such a world would still have regularities. They would just be different from the regularities that hold in our world.

And he proceeds to argue (in the same single-paragraph reply) that this differently regular world would still provide opportunities for "soul-making". (Sterba 2019, pp. 63–64).

However, this reply underestimates the problem of relatively frequent divine interventions. The problem is also understated by Almeida, who envisages that such interventions would be compatible with a world governed by laws of nature, but would be one where different positions and configurations of objects would generate different outcomes, apparently more benign in some regards than those in the actual world. (As will be seen later in this section, Almeida's interpretation of "differently regular" probably does not correspond to Sterba's.)

For relatively frequent divine interventions would involve counter-instances to most if not all the laws that operate in the actual world, the laws that govern gravity, light, sound, fluid dynamics, electro-magnetism, nuclear forces and the rest. Even if these laws are probabilistic, this frequency of interventions would still prove incompatible with them. Thus people about to become road casualties (for example, ones who had been pushed into a carriageway in front of a fast-moving vehicle) would be mysteriously lifted from the road and restored to a safe part of the adjacent sidewalk/pavement, or the oncoming vehicle would suddenly stall or mysteriously change direction without action on the part of the driver. Bullets in mid-air, already fired with murderous intent, would find their way back into the barrel of the gun they had been fired from. Bridges that had been negligently or fraudulently constructed with defective materials and were visibly beginning to collapse and bring the motorists and pedestrians using them to a certain death would mysteriously be reinstated and resume their

normal functioning. People who had been pushed from balconies would mysteriously cease to fall and be wafted up to a different, safer landing-place. People unjustly locked inside pitch-black underground dungeons would mysteriously receive light (coming from no discernible source) that showed them a previously non-existent tunnel to safety. And so on, including cases where fatal poisons already irretrievably administered would be counteracted by previously non-existent antidotes, and cases where the victims of fatal doses of deliberately deposited radioactivity would mysteriously be restored to health and vitality. The catalogue of cases would be lengthy and extend to infringements of a wide range of well-established natural laws.

Rather late on in his book, Sterba explicitly acknowledges that God's interventions, at least to prevent the more horrendous outcomes of natural evil, would have to take the form of miracles. Thus he writes:

> Of course, many of God's interventions would have to be miraculous, although they do not always have to appear as such. (Sterba 2019, p. 162)

His example of an intervention that might not appear to be miraculous is a cloudburst, that quenches enough of a forest fire to allow forest animals to escape being burned to death. But even if this might not appear miraculous, it would in fact constitute yet another infringement of natural regularities, and involve God superseding the laws that govern meteorology, and taking direct charge of the weather. It would also involve God subverting his or her own created order, and quite frequently too. Sterba certainly asserts that God's intervening to prevent horrendous evils would itself take place with law-like regularity, or "always" (Sterba 2019, p. 189). But such recurrent interventionism would not resemble the natural regularities discoverable by science; rather it would perpetually undermine them. Here doubts creep in about whether acting in this way can be expected of a good God, let alone expected as a requirement of God being good.

Even if there were some remaining natural regularities in such a world, there are large questions about whether a good God would create such a world of comparative chaos, lacking reliable regularities of the everyday kind, on which living creatures depend, and which have allowed them to evolve into the species that we find around us. So we should consider whether an all-good God, intent on making provision for living creatures and eventually for rational creatures, would or would not be likely to select for creation a world of natural regularities (without exceptions of the above-mentioned kinds), and whether he or she would be likely to select for creation a world with the regularities that obtain in the actual world.

With regard to the first of these questions, the answer appears to be affirmative. The actual laws of nature governing sound, light, gravity, heat, fluid pressures and the rest are the context of the evolution of living creatures, and it is hard to see how such creatures could have evolved in their absence. However, the central point is that some set of natural regularities would have to be selected if life was to be possible; and recent scientific findings suggest that there was a very narrow range of possibilities for cosmic constants such as that of gravity if the emergence of life was to be feasible. The universe seems to be "fine-tuned" for life, in terms not only of the positions of stars and planets but also of the operative laws of nature (see Attfield 2006, pp. 100–6, 120–23). Ward has argued cogently that just such a system of nature is the kind of system that a good God would be likely to select for creation (Ward 2007, pp. 68–73).

Besides, if a good God wanted to make provision for rational creatures, capable of understanding the world around them, and learning to act in ways generating largely predictable outcomes, then the creation of a regular world appears the only option. It is also an option that makes pursuits such as science possible, and thus a scientific understanding of the world in which we live.

But the second question is also important (and it is here that the present essay probably makes its most original contribution). Would a good God be likely to select for creation a world with the regularities that obtain in the actual world, or with different ones? This question is relevant, among other reasons, because Sterba writes of a world that would be regular, but differently regular, consistent

with interventions of the kind that he believes that an all-good God would instigate. However, as has been mentioned, the range of sets of natural regularities compatible with the emergence of life is a narrow one; and it is within this narrow range that anyone seeking to object that a different set of regularities from those of the actual world would be an improvement would have to locate a better set. Furthermore, this set would have to be better across the whole extent of space and the entire duration of time; granted that some kind of regularities are needed, it is not enough to depict laws that would benefit particular potential victims on a limited set of occasions only. The prospects for identifying a better set of natural regularities thus seem remote.

What is clear, however, is that there is probably no set of regularities that would facilitate the above-listed set of examples, comprising what we would regard as divine interventions to the laws of nature of the actual world. The mysterious eventualities listed above are unlikely to be consistent with any comprehensive set of natural laws or regularities applicable across space and time. And while there might remain some of the regularities of the actual world in a possible world in which these eventualities took place, a world with such eventualities is not compatible either with possibilities for "soul-making", or with the emergence of life, let alone with that of rational creatures, the kind of creatures eligible for "soul-making".

A world suited to the emergence of life and eventually of rational life would, by contrast, have characteristics such as universal regularities, alongside most of the other features that Ward lists in his chapter "The Integral Web" (Ward 2007, pp. 69–73). And as he remarks, persons "who have the same general nature that we do must be parts of a world of processes very similar to those of our own world" (Ward 2007, p. 68). Such a world is nothing like the possible world of the mysterious and seemingly miraculous "interventions" depicted above.

Here it should be added that the need for a world of living creatures (as well as for a world of rational creatures) to have such regularities is also relevant to the problem of natural evil, and thus of how an all-good God could create a world in which evils are present that do not derive from human action, but from natural forces and factors—evils such as disease, suffering and premature death. For evils of these kinds are, arguably, an inevitable outcome of such natural regularities. Sterba's argument eventually focuses on natural evils, and on God's non-prevention of the more significant kinds of such evils impacting both human beings and non-humans (Sterba 2019, pp. 157–80). I would reply, as before, that a good God would create a world governed by natural regularities, and not bring about infringements of them.

It might be suggested that a good God would constrain human freedom of action (as opposed to freedom of the will) and also opportunities for "soul-making" somewhat more than is actually the case, with a view to some degree of limitation of the current extent of suffering in the world. This suggestion, however, assumes that there is a trade-off between freedom of action and opportunities for "soul-making", on the one hand, and the reduction of suffering, on the other. But in fact, the very existence of rational and free creatures (as in the actual world) depends on the world being structured either by the natural regularities of the actual world, or of closely similar ones. If the world were structured with natural regularities outside this narrow range, or with none at all, free and rational creatures could not have evolved, and would not be present with their capacity to do the right freely. Hence, any relevantly differently structured world would not merely reduce freedom of action and opportunities for the development of moral character, but remove their possibility altogether, contrary to the spirit of the suggestion under consideration, which seeks to combine the presence of some limited amount of freedom of action and of opportunities for character development with reductions in current levels of suffering.

Sterba is able to reach his conclusions by regarding God as acting on a par with other moral agents. Thus he writes: "If it is always wrong for us to do actions of a certain sort, then it should always be wrong for God to do them as well." (Sterba 2019, p. 57). Certainly God is to be understood as a moral agent, a claim that Sterba well defends (Sterba 2019, pp. 111–17), but he writes as if God were the same kind of moral agent as ourselves, but just much more powerful. Instead, we need to bear in mind,

with Ward, that "God is good in the ways that are proper to the unique creator of all" (Ward 2007, p. 62). Predicates ascribed to God, such as goodness, make sense at the appropriate level, the level of creator. Accordingly, divine goodness does not involve (more or less frequent) interventions in the way that the goodness of a very powerful superhuman might well do. Divine goodness involves instead the creation of a regular world, a world that enables the emergence of life, of purposeful life, and of rational life, equipped with freedom of action and of creativity. We should not, as Taliaferro puts it, confuse or conflate "ethics within a world" with "the ethics of creating a world" (Taliaferro). Once this is recognised, the Free Will Defence can be understood against its broader context (as can the Irenaean theodicy), and, with this context understood, will turn out to comprise an adequate theodicy for moral evil. This same context, with its capacities for nurturing life, can also, I submit, be argued to comprise an adequate theodicy for natural evil.

## 6. Sterba's Argument from Rights

Sterba further argues that an all-good God would intervene to prevent not only the significant evil outcomes of free human action, but infringements of rights that a just political state would seek to uphold (Sterba 2019, 2020). Further, he appears to adopt a deontological approach to rights, suggesting that (subject to certain exceptions) there are foundational obligations of all moral agents to respect them, as and when they are able to do so.

That is not my own understanding of rights. On my understanding of rights, they are not morally fundamental, and do not generate fundamental obligations on the part of relevant moral agents. Rather, they derive from moral rules about the treatment of parties that we regard as rights-holders (whether human or non-human), rules that are morally mandatory in all but exceptional circumstances, because observance of these rules upholds the general good, but which admit of exceptions where the consequences of infringements of the rules outweigh the consequences of setting a precedent for infringing the rules. Rights, on this view, can be regarded as conclusions, rather than as basic premises (Attfield 2019, pp. 143–47).

Accordingly, if God's obligations were to be considered analogous to those of a very powerful human agent, Sterba's account of God's obligation to observe rights would already be subject to a scrutiny of whether infringements of the said rights were justifiable in terms of their consequences in the relevant circumstances. But God is not to be regarded as a moral agent on a par with a very powerful human agent; rather, God is to be regarded (let it be repeated) as (potentially) the creator of the material universe.

This is the context in which the question of whether human beings have rights against God would have to be asked. This question resembles the question considered by Paul (in his Epistle to the Romans) of whether a pot (or other vessel) could reasonably complain to the potter about its nature or function (Romans 9: 20–21). But I am no more committed to Paul's message here (implicitly that the pot has no such rights) than I am to the Pauline Principle, also based on the Epistle to the Romans, discussed above. Indeed, a good God is to be expected to recognise the moral rights of human and other rights-holders, where the implications of these rights for the obligations of human moral agents are concerned, and where no exceptions apply; and if God's will is to be understood as aligned to morality (as I join Sterba in holding), then a good God will generally favour rights being respected and matching obligations being observed by creaturely moral agents.

However, God is plausibly not obliged to create at all, let alone to create the actual world, or its creatures, or the moral right-holders and the moral agents within it. A good God may (and arguably has) nevertheless create(d) a regular world and make general provision for the flourishing of his or her creatures. But granted the evils to which the living creatures in a regular world are susceptible, and in particular the evils involved for such creatures as victims of immoral actions within the world's system of natural laws and regularities, there is a contradiction involved in holding that a good God would desire rational creatures, living in a regular universe, to exercise free will, with all that this state of affairs entails, and to develop mature characters, with all that this too entails, and that the same God

would intervene to prevent significant harms and suffering to creatures that become victims of the evil impacts of immoral creaturely actions. If the above argument about the need for a regular world, without frequent interventions to prevent the significant bad outcomes of immoral actions, stands up, then God cannot be obliged to intervene to prevent serious infringements of moral rights. If the argument works against God being obliged to prevent significant evils is successful in the first place, then it succeeds again in the context of interventions to prevent infringements of moral rights.

## 7. Compensation in an After-Life?

Sterba devotes some of his book to arguing that not even an afterlife with opportunities for "soul-making", followed by an unending period of bliss and beatitude, could compensate the victims of suffering and cruelty permitted by God to take place prior to, or culminating in, their death. His argument is that this permission on the part of God remains unfair to these victims, except where there is an "organic" link between their suffering and their post-mortem experience; and that in most cases there would be no such "organic" connection (Sterba 2019, pp. 35–45).

But it is far from clear what such an "organic" link amounts to, let alone whether it is as significant as Sterba suggests, as long as God grants former sufferers opportunities after death for "amendment of life" and/or (perhaps after that) perpetual bliss. As some theologians have recently maintained, such post-mortem redress could also be made available to animals whose lives were afflicted with suffering, unlikely as they are to discern any organic link between their sufferings and their subsequent happiness (Southgate 2008; Sollereder 2019).

It is also somewhat remarkable that Sterba in effect makes the absence of an "organic" link between this-worldly suffering and post-mortem compensation a sufficient condition of God's non-existence. For he holds that in the absence of such an "organic" link, God's permission of suffering is uncompensated, and since an all-good God would not permit uncompensated suffering, no such God can possibly exist. I have already contested the premise that an all-good God would under no circumstances permit uncompensated suffering, if its possibility were a necessary condition of the existence of the kind of regular world in which living creatures and eventually rational creatures with free will and capacities for character development could lead their lives (see the two previous sections). Yet a good God might also decide to grant post-mortem existence and a better life to creatures whose suffering would otherwise be uncompensated, and (in face of this possibility) the demand that such post-mortem existence and the kind of life that it made possible must be "organically related" to the sufferers' previous suffering appears too strong. Certainly this demand seems disproportionate to the claim that its non-satisfaction so conflicts with the nature of an all-good God that no such God can exist, even as a logical possibility.

Ward maintains that provision for some form of post-mortem existence is a necessary component of any satisfactory theodicy (Ward 2007, p. 72). I am not convinced about this claim, and take the view that a satisfactory theodicy can be found in the expanded version of the Free Will defence and of the Irenaean theodicy (expanded to include the requirement, for the existence of creatures to have free will and capacities for development of moral character, that the universe be structured by natural regularities close to the kind that our world actually exhibits). Yet it is worth tracing Ward's own account of post-mortem life, with a view to considering whether a victim of this-worldly suffering could be grateful for their life as a whole, rather than wish that he or she had never existed.

In considering this question, Ward discusses the story told by Ivan Karamazov within Dostoyevsky's novel *The Brothers Karamazov*.

> Ivan protests that if it is inevitable to torture a baby so that its unavenged tears are the foundation of final human happiness, such a world is not worth creating. Even an endless ecstasy for all people is not worth the torture of an innocent baby. (Ward 2007, p. 57)

Not even Ivan's pious brother, Alyosha, has a reply; he even agrees that he would not assist in the creation of such a world. But, as Ward proceeds to remark, Ivan's protest is built on the suppositions

that an alternative to the existence of the world described is possible, and that the torture is a means to the happiness of the rest of humanity. Ward suggests that these suppositions be replaced with a scenario such that

> there is no alternative, that universal happiness, for the baby as well as for the torturer, is only possible in a world where that torture may occur, and where, given many evil choices by finite wills, it does occur. (Ward 2007, pp. 57–58)

Further, we are asked to suppose that the torture is not a *means* to happiness, but a foreseeable but forbidden consequence of the existence of such a world. We further suppose that the torture is a possible consequence of the only sort of world in which happiness is possible for free finite creatures, and that "the baby can only have supreme happiness if it is born into a world in which it is so tortured" (Ward 2007, p. 58). Ward now suggests that the proper question to ask is:

> would that baby, after its terrible death, finding itself as a rational and mature agent in a world offering endless bliss, still say: 'I would rather never have existed'?

And his verdict runs: "In the perspective of endless bliss, that torture will soon diminish to the merest speck, an atom of misery lost in an eternity of bliss" (Ward 2007, p. 58).

Ward here accepts that the scenario with which he replaces that of Ivan Karamazov is a work of fiction. But it is, of course, itself a response to a fiction, indeed to a fiction within a fiction; and several of the features of Ward's scenario are later argued to be features of the actual world. Ward's readers might question whether the world of natural regularities, free will, and thus the possibility of torture, is also the only possible world allowing universal happiness for human beings, presumably after death. Yet if it is held that such happiness depends on the development of maturity of character, which in turn requires a world such as our own, then it is not unreasonable for Ward to include this supposition, as long as there is provision for attaining this maturity of character either before death or after it. Ward adds several qualifications and correctives to his response to Ivan Karamazov, which it is unnecessary to replicate here. For he appears to make the point that in the circumstances of a post-mortem life with prospects of bliss, the tortured baby, now a mature person, might reasonably be glad and grateful that he or she was born, as might other victims of suffering undergone in this life. He or she could reasonably be glad and grateful even if there were no "organic" connection between his or her suffering and his or her current life of bliss. Certainly, the former baby might need to go through a process of "soul-making" before attaining bliss, and might encounter in the course of it both challenges and new suffering (Hick 1977, pp. 350–52). Yet, even if so, her or his attainment of maturity and friendship with God could still make him or her glad to have lived.

Life after death would not, as in Plato's belief, be an implication of the natural immortality of human beings, but a gift of God, and, while a good God might confer it, God would not be obliged to do so. Yet the possibility that God does or will confer post-mortem existence and eventual bliss could be held to strengthen the case for God's goodness, and to weaken the case for the impossibility of the existence of an all-good God. The uncertainty of post-mortem existence could itself be regarded as an aspect of the cognitive distance that Hick claims to be indispensable for the development of human maturity; but the very possibility of bliss in a post-mortem existence is an additional ground for holding that the actual world, with all its evils, is the creation of a good God.

For Sterba, rejecting the possibility of satisfactory compensation for the victims of evil and suffering in this life serves an ancillary role; for the possibility of such compensation might buttress his theistic opponents' denials that God is obliged to intervene in this life to prevent the worst impacts of the immoral choices of free human agents, and of natural evils. The conclusion of this section is that this possibility does indeed reinforce the conclusion that God has no such obligation, an obligation which would in any case conflict with a good God's decision to create a world governed by natural regularities.

## 8. Conclusions

Sterba's case depends on the premise that a good God would have an obligation to intervene to prevent the worst impacts of immoral actions and choices, and also the worst impacts of natural evils. He reasons from the non-fulfilment of this purported obligation to the conclusion that the existence of the God of traditional theism is impossible.

But there is no sound basis for this premise. For this premise conflicts with the expanded version of the Free Will Defence, the version that includes the decision of a good God to create a world governed by natural regularities. It also conflicts with an expanded version of Hick's Irenaean theodicy, for opportunities for "soul-making" also depend on the world in which human action is situated being regular and partially predictable. Further, its claim that God is obliged to intervene to prevent the worst impacts of natural evils conflicts with what a good God would do when deciding to create a world without divine interventions and governed by natural regularities.

Sterba accepts that a good God would make provision for opportunities for "soul-making", but claims that God's failure to intervene in this world to prevent horrendous suffering shows that such provision is insufficiently made. Yet God's non-intervention turns out to be an implication of his or her decision to create a world that is regular and partly predictable; and provision for "soul-making" (or the development of a mature character, aligned with God's will) could also be provided in a post-mortem existence, if, as Sterba firmly contends, eternal bliss cannot be appropriately bestowed on people who have not undergone the kind of "soul-making" that he regards as a prerequisite (Sterba 2019, p. 37). (But this very requirement could well be an unnecessary abridgement of God's freedom and goodness; it may be true, but it is hardly reliable enough for an argument such as Sterba's for God's non-existence to depend on it, even in part.)

Accordingly, the claim that God has reasons (which can be partially grasped by human beings, but are not fully known to us) for permitting the world's evils appears after all to be tenable, and certainly to be logically possible. This in turn means that the existence of the world's evils (proposition B of the opening section of this essay) is implied by the conjunction of the existence of the God of traditional theism (proposition A of that section) and God having these reasons (a state of affairs corresponding to a proposition serving the role of C in that section). God's having these reasons is logically possible itself, and compatible with "A".

But this is enough to show that the existence of the God of traditional theism is compatible with the existence of all the evils of the actual world. I have argued elsewhere that God's existence is also probable, alongside the existence of these evils (Attfield 2006, 2017), but that is not the issue here. The main current conclusions are that Sterba's case for the incompatibility of the existence of God and of the evils of the actual world has not been made, and that God's existence and the world's evils are compatible after all.

**Funding:** This research received no external funding.

**Acknowledgments:** The author is grateful to Martin Warner and to Charles Taliaferro for their helpful comments on an earlier draft of this essay.

**Conflicts of Interest:** The author declares no conflict of interest.

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
