# Peer review of "Reconciling the God of Traditional Theism with the World’s Evils"

_religions, doi:10.3390/rel11100514_

Round 1
Reviewer 1 Report
This is an outstanding paper with a compelling case challenging Sterba's position.
This paper expertly addresses the classical problem of evil in light of a new version of the problem advanced by James Sterba of Notre Dame. Sterba has not been active in philosophy of religion, but focussed on moral and social-political theory. He has taken his work in that area to argue that an all good, omnipotent God should act like a good liberal state, protecting the vulnerable, etc. The author challenges this analogical reasoning and provides good reasons for thinking that the existence of God is compatible with world evils.
Author Response
(1) Reviewer 1 states that the author 'provides good reasons for thinking that the existence of God is compatible with world evils'.
I have now supplemented these reasons with a response to a suggestion offered by Reviewer 2. The newly added paragraph runs as follows:
It might be suggested that a good God would constrain human freedom of action (as opposed to freedom of the will) and also opportunities for ‘soul-making’ somewhat more than is actually the case, with a view to some degree of limitation of the current extent of suffering in the world. This suggestion, however, assumes that there is a trade-off between freedom of action and opportunities for ‘soul-making’ on the one hand, and reduction of suffering on the other. But in fact the very existence of rational and free creatures (as in the actual world) depends on the world being structured either by the natural regularities of the actual world, or by closely similar ones. If the world were structured with natural regularities outside this narrow range, or with none at all, free and rational creatures could not have evolved, and would not be present with their capacity to do the right freely. Hence any relevantly differently structured world would not merely reduce freedom of action and opportunities for the development of moral character, but remove their possibility altogether, contrary to the spirit of the suggestion under consideration, which seeks to combine the presence of some limited amount of freedom of action and of opportunities for character development with reductions in current levels of suffering.
Reviewer 2 Report
The article is very thorough, and while many of the arguments are not new, they deal with new criticism and with some (to me) minor new points, which means that it makes a new contribution to the field. The author may well specify it explicitly if there are specific new contributions to the theodicy discussion. In any case, the discussion covers all important points/objections and is worthy of publication.
Nevertheless, I have one main objection I would like the author to address. To explain what I have in mind, I start with a distinction between freedom of action and freedom of will, where the first is to be able to do what you will, and the second to be able to will what you will. In the discussion on free will the first is said to be a matter of having alternative possibilities, while the second is a matter of being the source of your choices. It seems possible that God could have created a world where we had less freedom of action to cause suffering and where the world we lived in caused less suffering. The author shows that in our universe it would have been chaotic, and it would have given less significant opportunities to develop moral character. Nevertheless, it still seems quite possible for an omnipotent God to create a (heavenlike) world with less freedom of action and less suffering. It would reduce our opportunity for soul-making, but many people would still think that it would be much better that God created a world with less suffering and less significant soul-making than to create a world with more suffering and more significant soul-making (this would especially be the view held by many of the victims of soul-making going wrong). This is then the objection I would like to hear the author reply to: A good and omnipotent God would have created a world with less suffering even if it implies less soul-making. If the author disagrees: why? How do you defend the value of soul-making being worth the price of suffering? (Or maybe the author does not disagree, but does not find it relevant. In any case, I would like to see the response.)
Author Response
1. Reviewer 2 suggests that I should find a way of conveying which part of my article is the most original.
I have done this by adding the following parenthesis at lines 343-344: (and it is here that the present essay probably makes its most original contribution)
2. Reviewer 2 makes a distinction between freedom of action and freedom of the will.
I have echoed this distinction at the beginning of the new paragraph that I have added in response to point 3, which I quote below.
3. Reviewer 3 suggests that a good God might partially limit (human) freedom of action so as to limit partially the extent of suffering in the world.
I have responded to this suggestion through inserting the following paragraph:
It might be suggested that a good God would constrain human freedom of action (as opposed to freedom of the will) and also opportunities for ‘soul-making’ somewhat more than is actually the case, with a view to some degree of limitation of the current extent of suffering in the world. This suggestion, however, assumes that there is a trade-off between freedom of action and opportunities for ‘soul-making’ on the one hand, and reduction of suffering on the other. But in fact the very existence of rational and free creatures (as in the actual world) depends on the world being structured either by the natural regularities of the actual world, or by closely similar ones. If the world were structured with natural regularities outside this narrow range, or with none at all, free and rational creatures could not have evolved, and would not be present with their capacity to do the right freely. Hence any relevantly differently structured world would not merely reduce freedom of action and opportunities for the development of moral character, but remove their possibility altogether, contrary to the spirit of the suggestion under consideration, which seeks to combine the presence of some limited amount of freedom of action and of opportunities for character development with reductions in current levels of suffering.